

# Evaluation of resistance to wheat stem rust and identification of resistance genes in wheat lines from Heilongjiang province

Qiujun Lin[*], Yue Gao[*], Xianxin Wu, Xinyu Ni, Rongzhen Chen, Yuanhu Xuan and Tianya Li

College of Plant Protection, Shenyang Agricultural University, Shenyang, China
[*] These authors contributed equally to this work.

## ABSTRACT

Wheat stem rust, caused by *Puccinia graminis* f. sp. *tritici*, (*Pgt*) is a devastating disease in wheat production. The disease has been effectively controlled since the 1970s due to the widespread use of the *Sr31* resistance gene. However, *Sr31* has lost its effectiveness following the emergence and spread of the Ug99 race variants. Therefore, there is an urgent global effort to identify new germplasm resources effective against those races. In this study, the resistance to *Pgt* of 95 wheat advance lines from Heilongjiang Province was evaluated using three predominant races of *Pgt*, 21C3CTTTM, 34C0MKGSM, and 34C3MTGQM, in China at the seedling and adult plant stage. The presence of 6 *Sr* genes (*Sr2*, *Sr24*, *Sr25*, *Sr26*, *Sr31*, and *Sr38*) was evaluated using linked molecular markers. The results showed that 86 (90.5%) wheat lines had plant stage resistance to all three races. Molecular marker analysis showed that 24 wheat lines likely carried *Sr38*, 15 wheat lines likely carried *Sr2*, 11 wheat lines likely carried *Sr31*, while none of the wheat lines carried *Sr24*, *Sr25*, or *Sr26*. Furthermore, six out of the 95 wheat lines tested carried both *Sr2* and *Sr38*, three contained both *Sr31* and *Sr38*, and two wheat lines contained both *Sr2* and *Sr31*. Wheat lines with known *Sr* genes may be used as donor parents for further breeding programs to provide resistance to stem rust.

## BACKGROUND

Wheat is the most important cereal grain in the world, contributing 20% of human caloric intake. Although more than 700 million tons of wheat are produced every year, food shortage has become a global problem, due to the rapid growth of the world population (*Food and Agricultural Organization of the United Nations (FAO), 2017*). In addition, the yield loss in the production of wheat caused by various wheat pathogens, including the wheat stem rust-causing fungus *Puccinia graminis* f. sp. *tritici* (*Pgt*), has accelerated this trend. Over the past 30 years, there has been no large-scale epidemic of wheat stem rust in China, because a large number of cultivars with effective stem rust resistance genes have been cultivated and popularized, which has played a vital role in controlling this disease

Corresponding author
Tianya Li, litianya11@syau.edu.cn

(*Li et al., 2016*). In addition, since the 1980s changes in crop layouts and in cultivation systems in the main overwintering regions that provide initial urediniospores of wheat stem rust (such as Fujian, Guangdong, and other provinces) have played a crucial role in controlling the overwintering of initial pathogens (*Li et al., 2016*). Nevertheless, wheat stem rust remains a long-standing threat to global wheat production security because of variations in the virulence of the *Pgt* population, and the ability of urediniospores to spread over long distances by wind (*Singh et al., 2015*). For example, the emergence of a new virulent race TTKSK (the well-known Ug99), was first identified in Uganda in 1998 (*Pretorius et al., 2000*), and was race-typed as race TTKSK in 2006 (*Jin et al., 2008*). The TTKSK is virulent against both *Sr31* and *Sr38* stem rust resistance genes, and this combination makes its virulence particularly significant. New variants of Ug99 with additional virulence on *Sr24* (TTKST), *Sr36* (TTTSK), and *SrTmp* (TTKTT and TTKTK) were identified since then (*Singh et al., 2011*). Moreover, new variants occurred with a reported loss in virulence, including *Sr30* (TTHST). Although races in the Ug99 group are virulent and there spread to over 13 countries poses a significant threat globally, they have not caused any epidemic, except in Kenya. While the International Maize and Wheat Improvement Center (CIMMYT) and the International Center for Agricultural Research in the Dry Areas (ICARDA) have established a global rust initiative to track and study Ug99 to prevent and control the disease on a global scale, in recent years new races of *Pgt* have emerged(e.g., TKTTF, TTRTF) causing a pandemic. Thus, stem rust is once again threatening worldwide wheat production (*Bhattacharya, 2017*; *Olivera et al., 2015*; *Olivera et al., 2019*).

Resistance breeding is the most effective, economical, and environmentally friendly strategy to control wheat stem rust. To date, at least 60 *Sr* genes have been identified in wheat and its wild relatives (*McIntosh et al., 2016*). While most confer race-specific resistance, some (including *Sr57*, *Sr58*, *Sr55*, and *Sr2*) do not confer race-specific resistance (*Juliana et al., 2017*) or high-temperature resistance (*Sr13*, *Sr21*) (*Chen et al., 2018*; *Zhang et al., 2017*). Therefore, it is of great significance to identify which resistance genes are present in which wheat cultivars and lines. This would guide wheat-resistance breeding and rational layout of resistance cultivars, avoiding the large-scale application of a single resistance gene and reducing the selection pressure of wheat cultivars to *Pgt*. The traditional identification approach of resistance genes is gene postulation according to the infection types (ITs) of the different stem rust resistance genes to known *Pgt* races. This strategy is easily affected by environmental conditions and is time-consuming, laborious, and complex (*Goutam et al., 2015*). In recent years, molecular marker technology has provided a new perspective on wheat disease management and has played an important role in molecular marker-assisted selection (MAS) breeding. One of the most important benefits of this technology is that markers are highly heritable and can be screened at the seedling stage. Due to the emergence and spread of Ug99, the mapping and application of molecular markers that were intricately linked with resistance genes of wheat stem rust has accelerated. So far, many molecular markers have been reported that are closely linked to wheat stem rust resistance genes (*Goutam et al., 2015*) many of which have been transformed into Simple Sequence Repeat (SSR), Sequence Characterized Amplified Region (SCAR), and

Sequence-Tagged Site (STS) markers that are widely used in wheat disease resistance molecular marker selection and breeding. *Haile et al. (2013)* used 31 markers linked to *Sr* genes to detect 58 tetraploid wheat in Ethiopia; The Ug99 resistant genes *Sr2*, *Sr22*, *Sr24*, *Sr36*, and *Sr46* were identified using markers linked with those genes in 99 Kazakh spring wheat (*Kokhmetova & Atishova, 2012*); *Mourad et al. (2019)* have confirmed the presence of stem rust resistance genes *Sr6*, *Sr31*, *Sr1RSAmigo*, *Sr24*, *Sr36*, *SrTmp*, *Sr7b*, *Sr9b*, and *Sr38* using gene-specific markers in Nebraska bread wheat germplasm. *Wu et al. (2014)* screened 139 Chinese wheat cultivars using markers linked with the Ug99 resistance genes *Sr22*, *Sr25*, *Sr26*, and *Sr28*. *Xu et al. (2017)* detected the resistance genes *Sr2*, *Sr24*, *Sr25*, *Sr26*, *Sr31*, and *Sr38* in 75 wheat cultivars in Gansu Province. Therefore, MAS is extremely helpful in identifying the tagged resistance genes which have been pyramided in one genotype.

Northeast China used to be an of the frequent occurrence of wheat stem rust, playing a key role in the large-scale epidemic of the disease. In history, there have been nine pandemics (1923, 1934, 1937, 1948, 1951, 1952, 1956, 1958, and 1964) in this area, and some years this even resulted in an almost total grain failure (*Wu & Huang, 1987*). In recent years, with the adjustment of agricultural structure, wheat production has mainly been distributed in Heilongjiang Province, where the annual planting area is nearly 300 thousand hectares. With the recent outbreak of wheat stem rust around the world, it is of great urgency to evaluate the resistance of wheat cultivars to *Pgt* and to clarify detailed knowledge of resistance genes present in wheat cultivars or lines. Therefore, we previously determined the level of resistance to *Pgt* of the 83 main production cultivars and the prevalence of *Sr2*, *Sr24*, *Sr25*, *Sr26*, *Sr31*, and *Sr38* in this region (*Li et al., 2016*; *Xu et al., 2017*). Based on these studies, we collected 95 advanced wheat lines to characterize the seedling and adult resistance levels to *Pgt*, and to identify the presence of *Sr* genes in those wheat lines using molecular markers. The results of our work will be important for developing potentially durable combinations of effective stem rust resistance genes in wheat cultivars.

## MATERIALS AND METHODS

### Plant and fungal materials

A total of 95 advanced wheat lines were collected from Heilongjiang Academy of Agricultural Sciences (Harbin, Jiusan, Hongxinglong, Heihe, Jiamusi) and Heilongjiang Bayi Agricultural University, covering the most important wheat-producing regions. Thirty-six monogenic lines with known stem rust resistance (*Sr*) genes, which were used in our study to test the virulence spectrum of *Pgt* and confirm the validity of these molecular markers, were provided by the Institute of Plant Immunity, Shenyang Agricultural University. The cultivar Little Club (LC) was used as a universal susceptible control. Three races (21C3CTTTM, 34C0MKGSM, and 34C3MTGQM) of *Pgt* with different virulence spectra (Table S1), which were used to evaluate the resistance level of the advanced wheat lines to *Pgt*, were identified using an international system of nomenclature for *Pgt* by the Institute of Plant Immunity, Shenyang Agricultural University (*Roelfs & Martens, 1988*; *Jin et al., 2008*).

## Seedling infection type (IT) assays

Seedling infection types (ITs) assays were conducted in duplicate in a greenhouse. The wheat lines were planted in a 12 cm diameter clay pot. The seeding ITs assays were carried out when the wheat seedlings grew to the two-leaf stage (one leaf and one sprout). First, the leaves were sprayed with a 0.05% Tween-20 solution using a handheld atomizer to form a water film on the leaves. Then, fresh urediniospores (1 g) and dried talc, mixed in a ratio of 1:20 (w/w), were inoculated on the seedlings. Following hydration in the dark for 16 h at 18 to $20°C$, the inoculated seedlings were transferred to a glass greenhouse with a temperature of $20 \pm 1°C$. When the universal susceptible control wheat line The cultivar LC was fully infected (14 days after inoculation), the seedling ITs were assessed according to the 0–4 scale described by *Stakman, Steward & Loegering (1962)*. According to this scale, 0–2 was classified as low infection type (resistant) while 3 and 4 were classified as high infection type (susceptible).

## Field stem rust evaluation

Resistance in adult plants was measured in three single-race nurseries in 2016 and 2017 at the experimental site of the College of Plant Protection, Shenyang Agricultural University (latitude 41°49′N, longitude 123°33′E, altitude 67 m). Seeds of each cultivar (line) were planted in double 1 m-rows, spaced 25 cm apart. The susceptibility control, LC, was planted perpendicular to all wheat cultivars between the double 1 m-rows. The various lines of wheat were inoculated at the green-and-jointing stage. Watering was achieved through sprinkling irrigation to ensure that the soil was fully humid before to inoculation, which was conducted in the evening. After spraying the leaves with a 0.05% Tween-20 aqueous solution, diluted urediniospores (urediniospores to talcum powder = 1:30 (w/w)) were sprayed as a powder onto the leaves for inoculation. A plastic cover maintained the moisture for 12–14 h. The infection responses (IRs) were assessed as immune ('I'), resistant ('R'), moderately resistant ('MR'), moderately susceptible ('MS'), or fully susceptible ('S'). Stem rust severity was assessed using a modified Cobb scale as described by *Roelfs, Singh & Saari (1992)*. When the LC was fully infected (14 days after inoculation), the first disease assessment was conducted which was repeated every 3 days. The highest IR and severity for each wheat variety was recorded.

## Molecular marker evaluation

DNA was extracted from the young leaves of 10-day old seedlings grown to the one-leaf stage, using a DNA extraction kit (http://www.sangon.com; China). Polymerase chain reactions (PCR) were carried out using an $S1000^{TM}$ Thermal Cycler in a volume of 25 μL, including 2 μL of 50 ng μL$^{-1}$ DNA, 1 μL of each primer (10 μmolL$^{-1}$), 2.5 μL of 10 × buffer (including Mg$^{2+}$ at a final concentration of 2.5 mM), 0.2 μL of *Taq* polymerase (5 U μL$^{-1}$), and 0.5 μL of deoxyribonucleoside triphosphates (10 mmolL$^{-1}$ each). PCR amplifications were done as previously reported (*Xu et al., 2017*). Six markers were used to identify the resistance genes 95 advanced wheat lines, and their effectiveness was confirmed using 36 monogenic lines with known *Sr* genes (Table S1). Primers were synthesized by Sangon Biotech (China) (Table 1), and PCR amplification conditions were as described

**Table 1** Molecular markers linked to resistance genes *Sr2, Sr24, Sr25, Sr26, Sr31,* and *Sr38* with their forward and backward primers.

| Tagged *Sr* gene | Primer | Fragment size (bp) | Primer sequence (5′–3′) |
|---|---|---|---|
| *Sr2* | *Xgwm533* | 120 | GTTGCTTTAGGGGAAAAGCC AAGGCGAATCAAACGGAATA |
| *Sr24* | *Sr24#12* | 500 | CACCCGTGACATGCTCGTA AACAGGAAATGAGCAACGATGT |
| *Sr25* | *Gb* | 130 | CATCCTTGGGGACCTC CCAGCTCGCATACATCCA |
| *Sr26* | *Sr26#43* | 207 | AATCGTCCACATTGGCTTCT CGCAACAAAATCATGCACTA |
| *Sr31* | *SCSS30.2 576* | 576 | GTCCGACAATACGAACGATT CCGACAATACGAACGCCTTG |
| *Sr38* | *VENTRIUP-LN2* | 259 | AGGGGCTACTGACCAAGGCT TGCAGCTACAGCAGTATGTACACAAAA |

in previous studies (*Xu et al., 2018*). Fragments of the targeted genes were separated by electrophoresis using 2% (w/v) agarose gels, stained with ethidium bromide, and observed under UV light.

# RESULTS

## Evaluation of wheat lines for stem rust resistance at the seedling stage

The ITs of 95 main wheat advance lines in Heilongjiang to the *Pgt* races 21C3CTTTM, 34C0MKGSM, and 34C3MTGQM at seedling stage are shown in Table 2. Three wheat cultivars Hong 09-1249, Gang 06-4, and Longfu 10-367 were susceptible (ITs 3-4) to all tested isolates at the seedling stage, accounting for 3.2% of the tested lines. Six wheat cultivars Jiusan 07-6205, Longfu09-1249, Jiusan 07-7371, Hong 09-558, Nong 11-2110, and Longfu 09-534 were resistant to one or two of the three tested races. The remaining 86 (90.5%) wheat cultivars were resistant to all tested isolates (Table 3).

## Evaluation of wheat lines for stem rust resistance at the adult plant stage

The IRs of 95 wheat lines to all tested isolates at the adult plant stage were determined during the 2017 and 2018 cropping seasons (Table 2). Based on the IRs, the tested wheat lines were classified into three groups. The first group (I) contained 21 (22.1%) wheat lines immune to all tested isolates displaying no visible symptoms (IT: 0) in two seasons. In the second group, 65 (68.4%) wheat lines showed MR-R (IT: 1, 1-, 1, 1+) with severity between 5%–50% to all tested isolates. In the third group, the remaining 9 (9.5%) wheat lines showed MS-S (IT: 3-, 3, 4) with severity between 60%–90% to all tested isolates (Table 4).

## Molecular identification

The adult plant resistance gene *Sr2*, originating from the tetraploid Yaroslav emmer, is located on chromosome arm 3BS. *Mago et al. (2011)* showed that a DNA marker *Xgwm533* is closely linked to this gene and that a 120 bp specific band could be amplified by PCR from

**Table 2  Infection types (ITs) on seedlings (2016) and infection responses (IRs) on adult plants (2016–2017), and amplification results for the known *Sr* genes by markers.**

| Line | Pidegree | 21C3CTTTM | | | 34C0MRGSM | | | 34C3MTGQM | | | *Sr* gene |
|---|---|---|---|---|---|---|---|---|---|---|---|
| | | ITs | IRS | | ITs | IRS | | ITs | IRS | | |
| | | 2016 | 2016 | 2017 | 2016 | 2016 | 2017 | 2016 | 2016 | 2017 | |
| Nongda09-1818 | Nongda03-2379/Nongda02-4339 | 0 | I | I | 0 | I | 5R | 1 | 10R | 5R | – |
| Nongda07-1328 | Longmai26/Nongda93-5233 | 1+ | 10R | I | ; | 5R | 10R | 0 | I | I | *38* |
| Nongda08-1639 | Nongda01-4576/Nongda01S-1197 | 0 | I | I | 0 | 5R | 5R | 0 | I | 5R | – |
| Nongda09-1884 | Longjian03-062/Nongda01-4551 | 0 | I | I | 0 | I | I | 0 | I | I | – |
| Jiusan07-6378 | Jiusan00-6709/Jiusan00-5006 | 0 | I | I | 0 | I | I | 0 | I | I | *31* |
| Jiusan07-6086 | Jiusan01F4-666/Kehan19 | 0 | 5R | 5R | 0 | I | I | 0 | I | I | *38* |
| Jiusan07-7106 | Jiusan01F3-889/Jianmai411 | ;1- | 10R | 10R | 0 | I | I | ;1 | 10R | 10R | – |
| Jiusan07-7409 | Beimai6/Jiusan01-6111 | 0 | I | I | 0 | I | I | 0 | I | I | *38* |
| Jiusan07-5221 | Nongda1662/Beimai6 | 0 | I | I | 0 | I | I | 0 | I | I | *38* |
| Jiusan07-7395 | Jiamai411/Jiusan99-6135 | 1 | 10R | 10R | 0 | 5R | 5R | 0 | I | I | – |
| Jiusan06-6203 | Jiusan01F3-889/Jiamai411 | 1 | 5R | 10R | 1+ | 10R | 20R | 2 | 10R | 30R | *31* |
| Jiusan07-6205 | Longmai30/Jiusan98-61249 | 3 | 30MS | 30S | 1 | 40R | 30R | 1+ | 30MR | 30R | – |
| Gang07-151 | Kenhong17/Gang94-441 | 0 | I | I | ; | 5R | I | ; | 5R | 5R | – |
| Gang09-558 | Longmai26/Gang94-445//Longfu990761 | 0 | I | I | ; | 20R | 20R | 0 | I | I | *38* |
| Pinzi66087 | CROC-1/A.SQ//2*OPATA/3/97-7293/4/94-4081 | 0 | I | I | ;1 | 20R | 30R | 0 | I | I | – |
| Pinzi96563 | N4/T.dicoccoides CWI19167 | 0 | I | 5R | 0 | I | I | ; | 5R | 5R | – |
| Pinzi86119 | Wangshuibai/Longmai26//Longmai19 | 0 | I | I | 0 | I | I | 0 | I | I | *2* |
| Pinzi96860 | Xinkehan9/SYN333//Longmai26 | 1 | 5R | 10R | ;1- | 5R | 20R | 1 | 20R | 20R | - |
| Pinzi96029 | ALTAR84/A.SQ(224) (R.F.)//97-7137/3/97-7146 | ; | 5R | 10R | 0 | I | I | 0 | I | I | – |
| Pinzi96864 | Xinkehan9/SYN333(R)//Longmai26 | ;1- | 10R | 10R | ;1- | 5R | 10R | 0 | I | I | - |
| Pinzi96248 | DR147/T.TIMOPHEVII CWI 17006//Longmai26 | 0 | I | I | 0 | I | I | 0 | I | I | *2* |
| Longfu05-0281 | 96RF6 1199qun/Longmai26 | 0 | I | I | 0 | I | I | 0 | I | I | *2, 38* |
| Longfu08-6247 | Fengshi001/Ke90-514 | 0 | I | I | 1 | 10R | 5R | 0 | I | I | – |
| Longfu08-586 | Kefeng10/Long00-0117 | 0 | I | I | ;1 | 5R | 10R | ; | 5R | 5R | *38* |
| Longfu09-1176 | Kang1396/Zhongyuan9//Kefeng10 | ; | 5R | 5R | 1- | 10R | 10R | ;1 | 10R | 20R | *38* |
| Longfu09-1249 | Kang151-4/Ke85-858//Keyin5 | 2 | 20R | 30R | 1 | 10R | 10R | 3- | 30MS | 30MS | – |
| Longfu09-0594 | Long00-0657/Shandong5-27 | 0 | I | I | 0 | I | I | 0 | I | I | – |
| Jiusan07-7371 | Jiusan01F3-889/Jiamai411 | 1+ | 30R | 30R | 3 | 50S | 60S | 4 | 70S | 80S | – |
| Jiusan07-6086 | Jiusan01F4-666/Kehan19 | 1− | I | I | 0 | I | I | 1+ | 5R | 10R | *31, 38* |
| Jiusan09u294 | JinK1638/Jiusan05yun40 | 1 | 10R | 20R | 0 | I | I | 1 | 20R | 20R | – |
| Hong09-556 | Gang03-441/M306 | 4 | 70S | 70S | 3 | 50S | 60S | 3- | 60S | 60S | – |
| Gang07-151 | Kenhong17/Gang94-441 | 0 | I | I | 0 | I | I | 0 | I | I | – |
| Gang09-557 | Longmai26/Gang94-445//M306 | 1 | 10R | 10R | 0 | I | I | 1 | 50R | 40MR | – |
| Gang06-4 | Beimai2/Yuan96-3 | 4 | 70S | 70S | 3 | 50S | 70S | 4 | 40S | 90S | – |
| Hong10-598 | Long01-1122/Nongda97-2829 | 0 | I | I | 0 | I | I | 0 | I | I | *2, 38* |
| Hong10-614 | Nongda00-1667/Long01-1237 | 0 | I | I | 0 | I | I | 2 | 20R | 10R | – |
| Hong10-595 | Long01-1122/Longmai00-4379 | 0 | I | I | 0 | I | I | 0 | I | I | – |
| Hong09-558 | Gang03-441/Longmmai990761 | 0 | I | I | 1- | 5R | 5R | 4 | 40S | 40S | – |
| Hong09-552 | Gang03-441/M306 | 2 | 10R | 10R | 0 | I | I | 1+ | 30MR | 30R | – |

**Table 2** (*continued*)

| Line | Pidegree | 21C3CTTTM | | | 34C0MRGSM | | | 34C3MTGQM | | | *Sr* gene |
|---|---|---|---|---|---|---|---|---|---|---|---|
| | | ITs | IRS | | ITs | IRS | | ITs | IRS | | |
| | | 2016 | 2016 | 2017 | 2016 | 2016 | 2017 | 2016 | 2016 | 2017 | |
| Hong10-600 | Long01-1122/Nongda97-2829 | 0 | I | I | ;1 | 20R | 20R | 1 | 10R | 20R | – |
| Nong10-0853 | Jiuyumai23/Le89-446//Long01-1237 | 0 | I | I | 0 | I | I | ; | 5R | 5R | – |
| Nong10-0313 | Long03F3-6519/Longfu20-378 | 1- | 30R | 30R | 0 | I | I | 1 | 10R | 30R | – |
| Nong10-0449 | Huapei3901/Jiusan62504//Longfu03Chanjianbeibao1 | 0 | I | I | 1 | I | I | 1 | 5R | 5R | *31* |
| Nong09-9702 | Longmai26/Long02-2165 | 0 | I | I | 0 | I | I | 1 | 10R | 30R | *38* |
| Nong10-0631 | Long04-4370/Long02-2309 | 0 | I | I | 0 | I | I | 0 | I | I | *2, 38* |
| Nong10-0333 | Long03F3-6519/Longfu20-378 | 0 | I | I | 0 | I | I | 1+ | 30R | 30R | – |
| Nong10-7767 | Long94-4081/Xiaobing32 | 0 | I | I | 0 | I | I | 0 | I | I | *31* |
| Nong10-0150 | Ke89-446/Jiamai6 | ;1 | 30R | 30R | 0 | I | 5R | 2 | 40MR | 20MR | – |
| Nong10-0632 | Long04-4370/Long02-2309 | ;1- | 10R | 5R | 2 | 30R | 20R | 2 | 30R | 30R | – |
| Nong11-1540 | (Huapei3901/Jiusan62504)F .1/Fancai | 0 | I | I | 1 | 20R | 20R | 1+ | 40R | 40R | – |
| Nong11-1530 | (Huapei3901/Jiusan62504)F .1/Hebai9413 | 0 | I | I | 1+ | 10R | 10R | 2 | 20R | 20R | – |
| Nong11-1393 | (Longmai26/Long98-8906)F .1/Long2003M8059-3 | ;1 | 10R | 10R | 0 | I | I | 1 | 10R | 20R | *2, 38* |
| Nong11-1664 | [(Longmai26/Zhouzhou9023)/Longmai29]F .1/Fancai | 0 | I | I | ;1 | 5R | 10R | 1+ | 30R | 30R | *2* |
| Nong11-1494 | (Long03-3152/Jiusan62504)F .1/Fancai | 0 | I | I | 0 | I | I | 2 | 40MR | 40MR | – |
| Nong08-8830 | 91-1178/Yemao | 0 | I | I | 0 | I | I | 0 | I | I | – |
| Nong11-1789 | 97-7215/95-3577//Longmai26 | 0 | I | I | 0 | I | I | 2 | 5R | 10R | *2, 38* |
| Nong10-0453 | Huapei3901/Jiusan62504//Longfu03Chanjianbeibao1 | 0 | I | I | 0 | I | I | 0 | I | I | *31* |
| Nong11H1336 | Jiusan99-5611/Kefeng6 | 0 | I | I | 0 | I | I | 2 | 20R | 20R | *2, 31* |
| Nong11za136 | Huapei3901/Jiusan62504//Longfu20-378 | 1+ | 20R | 10R | 1 | 20R | 20R | 2 | 30R | 20R | – |
| Nong10-0852 | Jiuyubai23/Ke89-446//Long01-1237 | 0 | I | I | 0 | I | I | 1 | 10R | 20R | – |
| Nong10-0149 | Ke89-446/Jiamai6 | 1+ | 30R | 30R | 1 | 30R | 40R | 1+ | 40R | 50R | – |
| Nong11-2097 | Ke90-513/Longmai30 | 1 | 10R | 10R | 0 | I | I | 1 | 5R | 5R | *31, 38* |
| Nong10-0995 | Ke93-387/Long99-9126//Long01-1237 | 0 | I | I | 0 | I | I | 1 | 10R | 5R | – |
| Nong06F6-5299 | Kefeng6/Liupeiti//83199/3/3901 | 2 | 30MR | 30MR | 1 | 20MR | 20MR | 1- | 20R | 20MR | – |
| Nong10-0716 | Long03-3152/Jiusan62504 | 0 | I | I | 0 | I | I | 0 | I | I | *38* |
| Nong11-1432 | Long03-3651/Longfu93-217//Longmai26 | 1 | 10R | 10R | 0 | I | 5R | 1 | 10R | 10R | *2* |
| Nong11-2110 | Long03-3675-1/Longfubeibaochanjian1 | 3 | 20MS | 20MS | 1 | 10MR | 20MR | 4 | 40MS | 40S | – |
| Nong10-0334 | Long03F3-6515-19/Longfu20-378 | 1 | 20R | 20R | ;1 | 5R | 10R | 2 | 10R | 40R | – |
| Nong10-0629 | Long04-4370/Long02-2309(Ke165-3/Longfu10) | 0 | I | I | 0 | I | I | 1 | 5R | 20R | *38* |
| Nong11-2294 | Long04-4370(Kefeng6/Long94-4081)/Long02-2309 | 0 | I | I | 0 | I | I | 1 | 10R | 20R | – |
| Nong11-2289 | Long04-4370/Long02-2309(Ke165-3/Longfu10) | 0 | I | I | 0 | I | I | 0 | I | I | *38* |
| Nong06-7721 | Long91-1131/Long94-4081 | ;1 | 5R | 10R | ; | 10R | 10R | 1 | 20R | 20R | – |
| Nong11-1062 | Long94-4081/Long97-7146(Longmai30) | ;1 | 5R | 20R | 1- | 20R | 20R | 1+ | 20R | 20R | *38* |
| Nong10-0509 | (Longfumai10/Longmai26)/Kefeng5//Long01F3-5050-1 | 0 | I | I | 0 | I | I | 1+ | 40MR | 40R | *2* |
| Nong10-0518 | Long99-6189-2/Kefeng5//Long01F3-5050-1 | 0 | I | I | 0 | I | I | 1 | 20R | 20R | – |
| Nong11-1056 | Longmai26/Kefeng4 | 0 | I | I | ; | 10R | 10R | 1 | 10R | 5R | *38* |
| Nong11-1017 | Longmai26/Lelao6 | 0 | I | I | ; | 5R | 5R | 1 | 30R | 30R | *2* |
| Nong11-1027 | Longmai26/(Kefeng5/Xiaobing33//Longmai26) | 0 | I | I | 0 | I | I | 0 | I | I | *2, 31* |
| Nong10-0070 | Longmai26/Longmai30 | 0 | I | I | 0 | I | I | ;1 | 10R | 5R | *38* |

| Line | Pidegree | 21C3CTTTM | | | 34C0MRGSM | | | 34C3MTGQM | | | Sr gene |
|---|---|---|---|---|---|---|---|---|---|---|---|
| | | ITs | IRS | | ITs | IRS | | ITs | IRS | | |
| | | 2016 | 2016 | 2017 | 2016 | 2016 | 2017 | 2016 | 2016 | 2017 | |
| Nong11-1074 | Longmai30/Kelao6 | 2 | 30R | 30R | 0 | I | I | 2+ | 50MR | 40MR | – |
| Nong11H1029-2 | Qi565-1*Long03-3651 | 0 | I | I | 0 | I | I | 1 | 30R | 30R | – |
| Nongda10-2001 | Nongda95-1743/Nongda93-5223 | 0 | I | I | 0 | I | I | 1 | 10R | 10R | – |
| Nongda10-1199 | Nongda04-2144/Nongda02-4541 | ;1 | 5R | 10R | 1- | 5R | 5R | 1 | 5R | 5R | – |
| Nongda07-1328 | Longmai26/Nongda93-5233 | ; | I | I | 0 | I | I | ; | I | I | 2, 38 |
| Longfu09-534 | (Kefeng10/Ke95RF-1750)F .0 | 0 | I | I | 0 | I | I | 3+ | 50MS | 40S | – |
| Longfu10K329 | Jiusankang151-6 Mutagenic Line | 0 | 5R | 5R | 1 | I | I | 1 | 10R | 10R | 31 |
| Longfu10-848 | Longfu02-12518/03K604 | 1 | 5R | 10R | ; | I | I | 1+ | 30R | 30R | – |
| Longfu10-367 | Long0657/Jiusan3U108 | 4 | 70S | 80S | 4 | 80S | 80S | 4 | 90S | 80S | – |
| Longfu10-797 | Gang03Jian912/01-4379 | 1 | 5R | 5R | 0 | I | I | ;1 | 30R | 30R | – |
| Longfu10-683 | Long6239-CH5R-2/Kefeng10 | ; | I | 10R | 0 | I | I | ;1 | 20MR | 20R | 2 |
| Longfu10-527 | Kefeng10/Ke96RF6-976//Ke95R498SP4 | 0 | I | I | 0 | I | I | 0 | I | I | 38 |
| Longfu11-243 | 04-711/Beiyin01-4 | 1- | 5R | 5R | 0 | I | I | ;1 | 30R | 30R | – |
| Longfu09-358 | (Ke88-418/Shandong95-9195)F .0 Mutagenic Line | 1 | 20R | 30R | 1 | 20R | 40R | 2 | 10R | 30R | – |
| Longfu08-6564 | Kefeng10 Mutagenic Line /Long00-0117 | 0 | I | I | 0 | I | I | 0 | I | I | 31, 38 |
| Longfu10-891 | Longfu01-4379/Kefeng9 | 1 | 10R | 10R | ; | 5R | 5R | 1+ | 30R | 30R | – |
| Little Club | – | 4 | 90S | 90S | 4 | 90S | 100S | 4 | 100S | 100S | – |

**Notes.**

[a]IT: infection types scored in the greenhouse seedling tests were based on a 0-to-4 scale (*Stakman, Steward & Loegering, 1962*) where ITs, 0, 1, or 2 were considered resistant and ITs 3 or 4 susceptible; and symbols + and –indicated slightly larger and smaller pustule sizes, respectively.

[b]IR: Infection responses were scored at the adult plant stage in the field tests following the descriptions of *Roelfs, Singh & Saari (1992)*, where I = immune, R = resistant, MR = moderately resistant, MS = moderately susceptible, and S = susceptible.

**Table 3  Resistant proportion of 95 advance wheat lines to three races of *P. graminis f. sp. tritici* at seedling stage.**

| Races | Susceptible | | Resistance | |
|---|---|---|---|---|
| | Number of lines | Percentage (%) | Number of lines | Lines (%) |
| 21C3CTTTM | 5 | 5.26 | 90 | 94.74 |
| 34C0MRGSM | 4 | 4.21 | 91 | 95.79 |
| 34C3MTGQM | 8 | 8.42 | 87 | 91.58 |
| All tested races | 9 | 9.4 | 86 | 90.5 |

**Table 4  Resistant proportion of 95 advance wheat lines to three races of *P. graminis* f. sp. *tritici* at adult stage.**

| Races | Immune | | Resistance-moderately resistance | | Moderately susceptible-susceptible | |
|---|---|---|---|---|---|---|
| | 2016 | 2017 | 2016 | 2017 | 2016 | 2017 |
| 21C3CTTTM | 56 (58.9)[a] | 55 (57.9) | 34 (35.8) | 35 (36.8) | 5 (5.3) | 5 (5.3) |
| 34C0MRGSM | 59 (62.1) | 57 (60.0) | 32 (33.7) | 34 (35.8) | 4 (4.2) | 4 (4.2) |
| 34C3MTGQM | 30 (31.6) | 29 (30.5) | 57 (60.0) | 58 (61.1) | 8 (8.4) | 8 (8.4) |
| All tested races | 21 (22.1) | 21 (22.1) | 65 (68.4) | 65 (68.4) | 9 (9.5) | 9 (9.5) |

**Notes.**

[a]56 (58.9): 56 = Number of wheat lines immute to tested race, 58.9=Percentage of immune wheat lines in total tested lines.

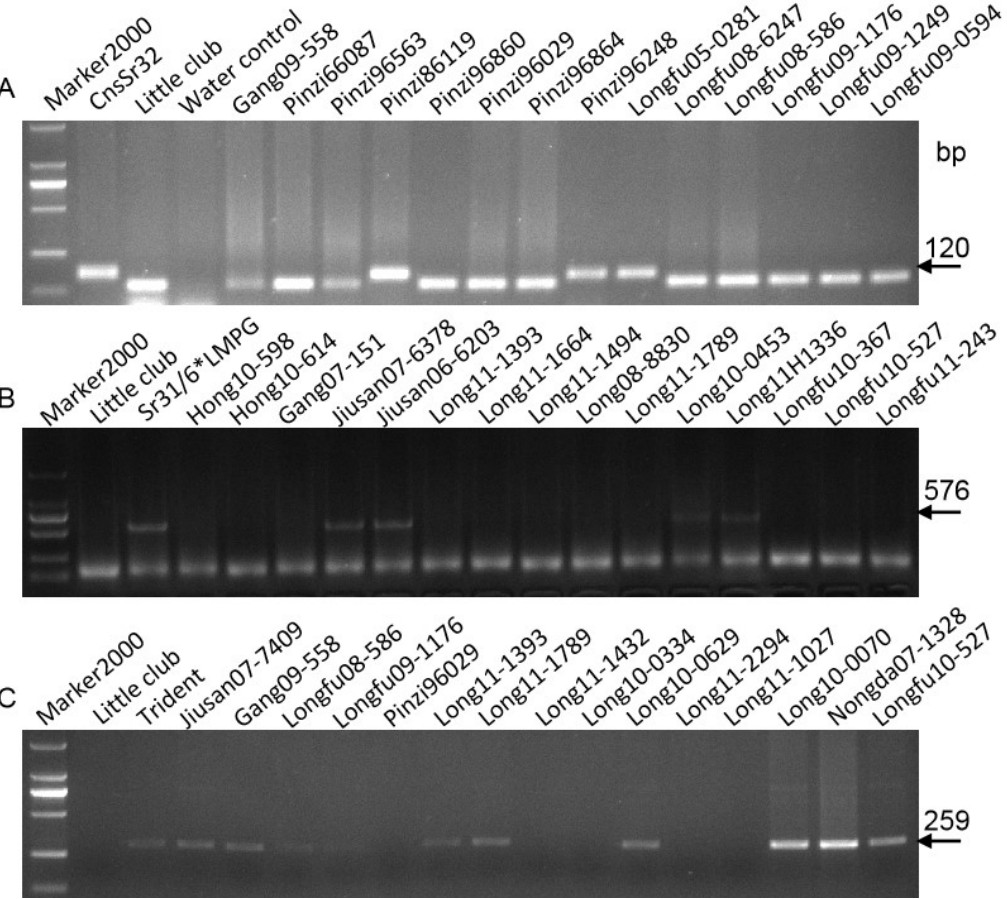

**Figure 1 Electrophoretograms of primers for different *Sr* genes.** The corresponding primers for *Sr2* (A), *Sr31* (B), and *Sr 38* (C) were used to screen Heilongjiang wheat lines.

wheat cultivars (lines) carrying this gene. This marker was used to determine the presence of *Sr2* in the 95 advance wheat lines from Heilongjiang Province. Fifteen wheat lines as well as the positive control line 'Hope' produced the 120 bp band (Fig. 1A), indicating that these wheat lines carry the resistance gene *Sr2*.

A specific molecular marker, *Sr24 #12*, was developed to detect the presence of the *Sr24* gene. A specific band of about 500 bp could be amplified by PCR in the wheat line LcSr24Ag, known to contain *Sr24*. The results showed that this fragment was only amplified in the positive control, LcSr24Ag, but not in the negative control LC or in any of the tested lines, indicating the likely absence of the *Sr24* gene in those wheat lines.

The resistance genes *Sr25* and *Sr26* are derived from *Thinopyrum elongatum*. These two genes provide good resistance to Ug99 and its variants. For this reason, the 95 wheat lines were subjected to PCR amplification with marker *Gb* (130 bp) linked with *Sr25* and with *Sr26#43* (207 bp) linked with *Sr26*. No specific fragments corresponding to these two primers were amplified in any of the tested wheat lines, except for their positive controls

Agatha/9*LMPG and Eagle, respectively, indicating the absence of those two genes in all 95 wheat lines.

Wheat stem rust gene *Sr31* originated from rye and has been deployed worldwide in many wheat cultivars. The molecular marker $SCSS30.2_{576}$, producing a 576 bp specific PCR fragment, was used to characterize the absence or presence of *Sr31*. Out of 95 wheat genotypes tested using $SCSS30.2_{576}$, the 576 bp fragment was identified in 11 wheat lines (Jiusan07-6378, Jiusan06-6203, Jiusan07-6086, Long10-0449, Long10-7767, Long10-0453, Long11H1336, Long11-2097, Long11-1027, Longfu10K329, and Longfu08-6564) as well as in the positive control Sr31/6*LMPG (Fig. 1B), indicating that those 11 advanced wheat lines carry the *Sr31* gene.

The *Sr38* gene, linked with leaf rust gene *Lr37* and stripe rust gene *Yr17*, originated from *Triticum ventricosum* and is located on a 2NS/2AS translocation. The 2NS-specific STS marker *VENTRIUP-LN2* was used to detect the presence of the gene cluster. *VENTRIUP-LN2* amplified a 259 bp band in the positive control and in 24 (25.3%) wheat lines (Fig. 1C, Table 2), confirming the presence of the *Sr38* gene in those 24 lines.

## DISCUSSION

The stem rust resistance gene *Sr2* that originated from tetraploid emmer wheat (*Triticum dicoccum Schronk*) has provided durable broad-spectrum, adult-plant resistance to wheat stem rust (*Singh et al., 2011*). The gene is located on chromosome 3BS and causes resistance to many *Pgt* races (*Hayden, Kuchel & Chalmers, 2004*). It was introduced into North America and the CIMMYT wheat breeding program in 1925. Since then, it has been widely deployed in many countries, including China. This gene was combined with *Sr33* in production for nearly 70 years and remained resistant (*Periyannan et al., 2013*). Here, using a molecular marker, we identified fifteen wheat lines that carry *Sr2*, and that display all-stage resistance to the tested *Pgt* races 21C3CTTTM, 34C0MRGSM, and 34C3MTGQM. However, the Hope line, which carries a single *Sr2* gene, was susceptible to these *Pgt* races 21C3CTTTM, 34C0MRGSM, and 34C3MTGQM at the seedling stage, therefore the 15 resistant wheat lines may contain another unknown resistance gene that confers resistance to the above three races at the seedling stage. Therefore, these resistant materials can be purposefully used to improve the resistance level of Heilongjiang wheat varieties to Chinese *Pgt* and Ug99 in future disease resistance breeding.

The *Sr24* gene, originating from *Thinopyrum ponticum* and located on 3DL of the wheat chromosome, is widely used in wheat production worldwide. The gene does confer resistance to Ug99 (TTKSK), but it was added to a North American system of nomenclature after a new variant of Ug99 (TTKST) gained virulence on *Sr24*. Although *Sr24* did not provide resistance to some variants of Ug99, it provided excellent resistance to most Chinese races and to the new races TKTTF and TTRTF that caused disease epidemics in Ethiopia and Italy in 2014 and 2016, respectively (*Olivera et al., 2019*). In a previous study, the molecular marker *Sr24#12* was used to screen wheat cultivars from Heilongjiang province. Unexpectedly, no wheat varieties that might contain this gene were found in 83 tested wheat materials (*Li et al., 2019*). As we expected, in our current work no wheat lines

that contain the gene were found in 95 tested wheat lines, in agreement with our previous study indicating main commercial wheat cultivars do not carry this gene.

The *Sr25* and *Sr26* genes were derived from *Thinopyrum ponticum*. These two genes provided excellent resistance to Ug99 strains, TKTTF and TTTRF and to all races of *Pgt* that are found in China (*Li et al., 2019*). Recently, with the diversification of breeding methods, considering their excellent ability to provide resistance to Ug99 and its variants, wheat breeders in various countries began to use these two genes to improve the resistance to stem rust of wheat. Since *Sr25* is a temperature-sensitive gene, its resistance is affected by the growth period and temperature (*Friebe et al., 1994*). The resistance at the seedling stage is higher than at the adult stage, and the plants are more susceptible at high temperatures (*Jain et al., 2009*). Research has shown that *Sr25* is almost absent from wheat varieties in China, and our results confirm this (*Li et al., 2016*). *Sr26* is mainly applied to wheat breeding in Australia, and it is seldom used in China. No wheat lines containing *Sr26* were found among the tested varieties in this study. Combining our results with those from previous reports, *Sr26* was not found in nearly 400 wheat materials collected from different regions of China (*Li et al., 2016*; *Li et al., 2019*; *Xu et al., 2017*; *Xu et al., 2018*). Therefore, we would suggest that the introduction of this gene into wheat breeding in China would enrich the diversity of resistance sources of its wheat varieties.

The *Sr31* gene is one of the most widely used stem rust resistance genes in wheat breeding in the world. It is located on 1BL/1RS chromosome and was first transferred from 'Petkus' rye to bread wheat (*Mago et al., 2002*). In the 1960s, China began to introduce 'Soviet Union' and 'Romania' wheat strains containing *Sr31* (*Jiang et al., 2007*). Since then, this gene has been widely used in wheat breeding in China, and the cultivated area of wheat varieties carrying this gene accounts for more than 60%. Although *Sr31* has "lost" its effectiveness to Ug99 races, it has always provided excellent resistance to all domestic stem rust isolates in China's wheat production. Knowing the distribution of this gene in domestic cultivars is of practical significance for monitoring for Ug99 and preventing the occurrence of stem rust in China. In this study, the *Sr31*-linked marker $SCSS30.2_{576}$ was used to detect the distribution of this gene in 95 wheat lines from Heilongjiang province, and pedigree analysis revealed that 11 of those wheat lines carried *Sr31*. The characterization of the resistance of these wheat lines to three races of *Pgt* also supported this result, since all of these wheat lines produced low ITs (0 to 2) at the seedling stage, and were immune (I), resistant (R), or moderately resistant (MR) at the adult-plant stage with relatively low severity (<30%). Thus, our results suggest that there are relatively few wheat varieties containing *Sr31* in Heilongjiang province, less than in other provinces in China (*Cao et al., 2019*; *Xu et al., 2017*; *Xu et al., 2018*).

The *Sr38* gene originated from *Aegilops ventricosa* L. It was first transferred into the winter wheat variety 'VPM1' and is closely related to the stripe rust resistance gene *Yr17* and the leaf rust resistance gene *Lr37* in wheat (*Bariana et al., 1993*). The *Yr17-Lr37-Sr38* gene cluster is used globally in wheat production since it provides excellent combined resistance to stripe rust, leaf rust, and stem rust of wheat. In this study, specific fragments were amplified in 19 wheat lines, indicating that these lines may contain *Sr38*. In addition, the ITs and IRs also support this result, exhibiting an IT of 0 to 2 at the seedling stage while

they were resistant (R) to immune (I) at the adult-plant stage to 3 tested *Pgt* races. Similar to *Sr31*, *Sr38* has also "lost" its ability to provide resistance to the Ug99 races, but no *Pgt* isolate can overcome this resistance in China. Therefore, *Sr38* will still play a role in the prevention and control of wheat stem rust, but Ug99-resistant genes should be aggregated in breeding to improve the resistance level of Chinese wheat cultivars to this disease.

Our results also showed that the wheat lines from Heilongjiang province displayed good resistance to three *Pgt* races. Of the 95 wheat lines tested, 86 (90.5%) not only had good resistance to the races 21C3CTTTM, 34C0MKGSM, and 34C3MTGQM at the seedling stage, but also showed good resistance to these three races in the resistance evaluation of two consecutive years at the adult stage with low severity (<30%). Therefore, those 86 wheat lines have all-stage resistance to the tested races. This may be related to the fact that resistance to *Pgt* is a breeding goal of wheat lines, and wheat cultivars approved in Heilongjiang province must be resistant to wheat stem rust. All wheat lines are screened with the predominant race group 21C3 and the sub-dominant race group 34 by the Plant Immunity Laboratory of Shenyang Agricultural University at the field nursery before registration, and only wheat lines with medium resistance or above can be registered as new varieties through a variety examination and approval. From the results of molecular detection, the wheat lines contain abundant resistant material including the broad-spectrum stem rust resistance genes *Sr2* as well as *Sr31* and *Sr38* that provide resistance to all wheat stem rust races occurring in China. Although these genes have been used to protect wheat from stem rust for many years and are still effective in China, new virulence to these genes is becoming more frequent and they are not completely effective anymore. Therefore, more *Sr* resistance genes (especially against Ug99) need to be evaluated using the molecular markers. This will give breeders a better overview of how diverse current wheat breeding material is in terms of stem rust resistance.

## CONCLUSION

The breeding of resistant cultivars is the most cost-effective and eco-friendly strategy to protect wheat from wheat stem rust. In this study, resistance to *Pgt* of 95 advanced wheat lines from Heilongjiang Province was evaluated at the seedling and adult plant stage using three predominant races of *Pgt* in China, including 21C3CTTTM, 34C0MKGSM, and 34C3MTGQM. Overall, the resistance level of wheat lines to wheat stem rust was strong in Heilongjiang Province. Based on these results, the presence of *Sr2*, *Sr24*, *Sr25*, *Sr26*, *Sr31*, and *Sr38* genes in these lines was detected using gene specific DNA markers. The results showed that 42 of the tested wheat lines carry one of these genes. This information can be used in future wheat-breeding strategies for obtaining stem rust resistance.

## ACKNOWLEDGEMENTS

We appreciate very much Weifu Song (Heilongjiang Academy of Agricultural Sciences), Qingjie Song (Heilongjiang Academy of Agricultural Sciences), Hongji Zhang (Heilongjiang Academy of Agricultural Sciences), Yantai Guo (Heilongjiang Land Reclamation Bureau Jiu San Institute of Agricultural Sciences), Bo Zhang (Heilongjiang Land Reclamation Bureau

Hongxinglong Institute of Wheat Research), Wang Lu (Heilongjiang Bayi Agricultural University) for providing the wheat lines and their pedigrees.

### Funding
This study was supported by the Natural Science Foundation of Liaoning Province (2020-MS-204) and the National Natural Science Foundation of China (No. 31701738). The funders had no role in study design, data collection and analysis, decision to publish, or preparation of the manuscript.

### Grant Disclosures
The following grant information was disclosed by the authors:
Natural Science Foundation of Liaoning Province (2020-MS-204).
National Natural Science Foundation of China: 31701738.

### Competing Interests
The authors declare there are no competing interests.

### Author Contributions
- Qiujun Lin conceived and designed the experiments, performed the experiments, prepared figures and/or tables, and approved the final draft.
- Yue Gao and Xianxin Wu performed the experiments, prepared figures and/or tables, and approved the final draft.
- Xinyu Ni, Rongzhen Chen and Yuanhu Xuan analyzed the data, authored or reviewed drafts of the paper, and approved the final draft.
- Tianya Li conceived and designed the experiments, prepared figures and/or tables, authored or reviewed drafts of the paper, and approved the final draft.

### Data Availability
Raw data is available in the Supplementary Files.

### Supplemental Information
Supplemental information for this article can be found online at http://dx.doi.org/10.7717/peerj.10580#supplemental-information.

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
