# Peer review of "Evaluation of resistance to wheat stem rust and identification of resistance genes in wheat lines from Heilongjiang province"

_PeerJ, doi:10.7717/peerj.10580_

## Round 0.1 · original submission · Major Revisions

In view of these reviewers and my own reading of the manuscript, your manuscript needs substantial revisions to address the concerns of the reviewers. Reviewer 1 has serious concerns about the experimental designs, data analysis possibly correlation analysis between phenotypic and molecular data), interpretation, and biological significance of your research. Reviewer 2 felt that the experiment was well done but that there are several problems with the paper that need to be addressed. In addition, I felt that the manuscript needs a fair bit of language and grammatical editing. Once you revise and resubmit the manuscript, I will look over the manuscript and either make a decision directly or send it out for further review. I have made this decision because I believe that the paper is of significant value but will require substantial rewriting prior to acceptance of the manuscript for publication in PeerJ.

·

Basic reporting

Well-written manuscript with suffiencient background and context provided.
Needs additional work on Results and Discussion sections.

Experimental design

Overall, the experiments are well-conducted, and the results are consistent with the experimental design proposed.

Validity of the findings

Please, read general comments for author.

Additional comments

October 6, 2020

Dear Editor of PeerJ journal,

Thank you so much for referring me the manuscript #52649, entitled ‘Evaluation of resistance to wheat stem rust and identification of resistance genes in wheat lines’. This research addresses a relevant topic, which is the understanding of the basis of stem rust resistance in breeding materials. Getting to know what genes are present in a breeding program is crucial for deciding the breeding strategies to combine effective genes to guarantee long-lasting resistance.

This research includes stem rust phenotyping at the seedling and adult stages against three Chinese races of the stem rust pathogen, and a molecular evaluation for the presence of six Sr genes in 95 breeding lines. Overall, the experiments are well-conducted, and the results are consistent with the experimental design proposed. However, I think this manuscript needs more elaboration and additional interpretation of the results that will definitely enrich both the Results and Discussion sections.

I am including here the major points that were missing or need to be reviewed:

1) A strong emphasis was put in this manuscript on finding resistance to Ug99. Based on your evaluation, you cannot conclude that your resistance will be effective against Ug99 races. The races you used here are different from Ug99 in terms of virulence. None of the three races you tested are virulent on Sr31 or Sr38 (critical virulences in Ug99 races). You can only conclude that your genes will be effective against Ug99 races if you evaluate your germplasm against these races. So, you need to emphasize that although these lines proved to have good resistance to the three races you used, their effectiveness against Ug99 races will be only determine once they get evaluated against Ug99 or proven by molecular markers that they carry genes that are effective against Ug99. If your original objective was to find lines that are resistant to Ug99 races, unless you run markers for additional genes that are effective to Ug99, you will not be able to address it.

2) There is an important missing point in relation to Sr24. If you see the three races you used in your study, all of them are virulent to Sr24. So, this gene is truly not a good option in your province. Maybe, the reason why you don't have Sr24 in your materials is that it is not a good gene in your province and was selected against by the virulent races. In your discussion (lines 216-217) you mentioned that Sr24 provides resistance to most Chinese races, but the three races you used in your research are virulent on Sr24. If you add a race avirulent on Sr24 in your seedling evaluations, you would be able to postulate this gene easily.

3) No discussion was made about the narrow basis of stem rust resistance in these materials. Only three Sr genes were identified with molecular markers: Sr2, Sr31, Sr38. And although these have been relevant genes used to protect wheat from stem rust for many years and are still effective in China, now virulence to these genes is becoming more frequent and are not that effective anymore. Need to add in the discussion that more genes need to be evaluated through the use of molecular markers. Will give you a better picture of how diverse your breeding material in terms of stem rust resistance is. Based on the phenotypes observed it does not seem to be highly diverse.

4) I was expecting to read what will come next after this research. Recommendations from this group about how to continue this research to better understand the basis of stem rust resistance in these highly resistant lines. Running markers for additional Sr genes? What criteria to select lines for stem rust improvement? Pyramiding genes (Sr2, 31, and 38 to protect from Chinese races + additional genes to protect from Ug99??)

5) No analysis was made to correlate the phenotypes and the markers data. Do the lines carrying Sr31 or Sr38 (or a combination of both) perform better than the lines that do not have them? Same for Sr2.



Additional edits:


Lots of missing citations in the Discussion section (I highlighted them).

Review the table's numbers. Not matching between tables number and manuscript body.

Title: I would add the origin of these lines (Heilongjiang province) at the end of the title.

Line 18: change ‘detected’ to ‘evaluated’.

Line 33: add ‘effective’ before ‘stem rust resistance genes’.

Lines 40-44: Needs a better description of the Ug99 race group, and how new variants in the group has emerged. Ug99 was first identified in Uganda in 1998 (Pretorius et al, 2000), and was race-typed as race TTKSK in 2006. Race TTKSK is virulent on both Sr31 and Sr38, making this virulence combination so significant. New variants of Ug99 with additional virulence on Sr24 (TTKST), Sr36 (TTTSK), and SrTmp (TTKTT and TTKTK) were identified since then (see Newcomb et al., 2016). But also, there have been new variants occurred that report a loss in virulence, like on Sr30 (TTHST). Although races in the Ug99 group are virulent and a big threat, they have not caused any epidemic, except for Kenya. The race group has been spreading, reaching 13 or more countries, but no epidemics were reported due to Ug99 other than big losses in Kenya.

Line 48: race TTTTF was wrongly called in Bhattacharya’s paper. The correct name of the race is TTRTF (See Patpour et al., 2017 and Olivera et al., 2019).

Line 54: Sr13 and Sr21 are not APR genes. They are both affected by temperature, but they are both all-stage resistance genes.

Line 75: change ‘Sr1RS’ for ‘Sr1RSAmigo’.

Line 83: Can you please elaborate more about when these pandemics occurred? Are they recent pandemics? Happened a long time ago?

Lines 105-106: Is it the virulence spectrum the only criteria for the selection of these races? Are these races prevalent in China?

Line 107: what international system of nomenclature are you referring to? I believe it is the North American system based on the race name. Please, you need to cite this.

Line 110: I don't think it is necessary to mention who did this evaluation. Please, eliminate the sentence.

Line 118: change ‘LC’ to “Little Club’

Line 119: change ‘investigated and recorded’ to ‘assessed’

Line 122: Field stem rust evaluation section. There is important information that is omitted here. Did you run three independent single-race nurseries? Or you run only one nursery and you bulk the three races for the inoculum? Based on your Table 2, seems like you had three independent races. This needs to be clearly stated in this section.

Lines 126-131: When you describe the field evaluations, it is not clear how you used the susceptible control LC. Did you use it as an inoculum spreader row? You only mentioned here that you inoculate your wheat lines, but nothing is mentioned about inoculating the susceptible control.

Line 137: I recommend changing the title of this section. It can be: Molecular markers evaluation. Please, mention that you run the six markers in your 95 advanced wheat lines.

Line 152-156: You mentioned here that 9 lines were susceptible to all races and the remaining ones resistant to all races at the seedling stage. This is not what Table 2 is showing. Lines like Jiusan07-6205, Longfu09-1249, Jiusan07-7371, Hong09-558, Nong11-2110, Longfu09-534, Longfu10-848, all of them show race specificity in the seedling responses. Please, check that and elaborate more in this results section.

Line 159: Before you said that the field evaluations were in 2016 and 2017. Please check. Results are presented in table 2 instead of 1, please check.

Lines 159-164: nothing is mentioned about the race specificity in the field response (same as in seedling).

Lines 165-166: Please, clarify what you mention in this sentence. Results appear to correlate very well in both years for each race.

Line 167: ‘Molecular identification’ section. This is a results section. Here, you are presenting a lot of information about each marker that does not fit with this section. All the information describing the markers should go in the Materials and Methods or in the Discussion section.
Also, nothing is mentioned about the lines that carry multiple resistance genes.

Line 199: Review the sentence, it is not correctly written.

Line 200-201: You need a citation for this sentence.

Line 213-215: This sentence needs to be revised. Yes, Sr24 confers resistance to Ug99 (TTKSK), but was added in the differential set after a new variant of Ug99 (TTKST) gained virulence on Sr24.

Lines 216-217: The race that caused the epidemic in Italy is race TTRTF. Please, add citations to this sentence.

Lines 228-230: Please, add citations to these sentences.

Lines 238-239: Please, add a citation to this sentence.


Table 2: For two lines (Pinzi96860 andPinzi96864) you wrote that Sr32 was confirmed with markers. There should be a typo, please check.

Supplemental Table 1.
It seems like the three races used are named based on the North American differential set and nomenclature system. When you present the data, instead of listing the genes in number order, it is clearer if you order them based on the 20 differential sets (Sr5, Sr21, Sr9e, Sr7b … etc.). List first the 20 genes in the differential set and then continue with the additional genes you evaluated.

Reviewer 2 ·

Basic reporting

No comment

Experimental design

No comment

Validity of the findings

No comment

Additional comments

This manuscript entitled “Evaluation of resistance to wheat stem rust and identification of resistance genes in wheat lines” provides essential information on the stem rust resistance available in Heilongjiang Province. The manuscript is well written in terms of Introduction, research questions, experimental design, and data analysis. It is understood that resistance is available against the predominant local races of Puccinia graminis f. sp. tritici (Pgt), however, it would be nice to see some discussion on strategies to increase resistance sources to Ug99 strains.

I also see need of careful revision in some parts of the manuscript, some of which have been listed below.

“International Center for Agricultural Research in the Dream Areas (ICARDA)” change to “International Center for Agricultural Research in the Dry Areas (ICARDA)”

“some confer non-specific resistance” change to “some confer race non-specific resistance:”.

I would suggest using “combined” instead of “polymerized”.

Some corrections are needed in References.

Change “Bhattacharya S. 2007” to “Bhattacharya S. 2017”

In figure 1, the first lane should be changed to “Marker2000”.

Please check if the Table numbers match with the text.

Please take a look at the edits in the PDF.

Thank you.

Annotated reviews are not available for download in order to protect the identity of reviewers who chose to remain anonymous.

---

## Round 0.2 · accepted · Accept

Thank you for submitting your revised manuscript (PeerJ submission 52649). I reviewed it and made minor edits to the pdf file. I am pleased to inform you that the review process of your manuscript has now been completed and accepted for publication.

I look forward to receiving the new submission of your manuscripts to PeerJ in the future.

Thank you.

Tika Adhikari